# Ultrasound Can Be Usefully Integrated with the Clinical Assessment of Nail and Enthesis Involvement in Psoriasis and Psoriatic Arthritis

**DOI:** 10.3390/jcm11216296

**Published:** 2022-10-26

**Authors:** Yu-Shin Huang, Yu-Huei Huang, Chiung-Hung Lin, Chang-Fu Kuo, Yun-Ju Huang

**Affiliations:** 1School of Medicine, Chang Gung University, Taoyuan City 33305, Taiwan; 2Department of Dermatology, Chang Gung Memorial Hospital, Taoyuan City 33305, Taiwan; 3Department of Thoracic Medicine, Chang Gung Memorial Hospital, Taoyuan City 33305, Taiwan; 4Department of Rheumatology, Allergy and Immunology, Chang Gung Memorial Hospital, Taoyuan City 33305, Taiwan; 5Center for Artificial Intelligence in Medicine, Chang Gung Memorial Hospital, Taoyuan City 33305, Taiwan

**Keywords:** Glasgow Ultrasound Enthesitis Scoring System, nail, psoriasis, psoriatic arthritis, ultrasonography

## Abstract

**Objectives:** This study aimed to examine and compare the findings of nail and enthesis ultrasonography in patients with psoriasis and psoriatic arthritis. **Methods:** We identified 154 patients with psoriatic arthritis and 35 patients with psoriasis who were treated at Chang Gung Memorial Hospital, Taiwan, between September 2018 and January 2019. **Results:** There were significant differences in the Nail Psoriasis Severity Index scores and Glasgow Ultrasound Enthesitis Scoring System scores between patients with psoriasis and those with psoriatic arthritis. B-mode ultrasonography revealed that onychopathic changes were more common in the psoriasis group. The psoriatic arthritis group showed a higher proportion of lower-limb enthesopathy, with significant differences in distal patellar ligament thickness and Achilles tendon thickness. **Conclusion:** The findings of nail ultrasonography were more severe in psoriasis cases, and the ultrasonographic findings of enthesopathy of the lower limb were more severe in cases of psoriatic arthritis.

## 1. Introduction

Psoriatic disease is a noncontagious, multisystem autoimmune disease with predominantly skin and joint involvement [1]. Psoriasis (PsO) manifests with scaly erythematous plaques usually affecting extensor surfaces of the elbows and knees, and sometimes involves other parts of the body [2]. Psoriatic arthritis (PsA) is a chronic multifactorial inflammatory musculoskeletal disease, usually associated with PsO [3]. PsA affects similar diverse organ systems involving the skin, nails, enthesis, and axial and peripheral joints [4]. Diagnosis of PsA is difficult due to variable clinical signs. Nevertheless, classification systems such as the Classification for Psoriatic Arthritis (CASPAR) criteria, and several other screening tools, can facilitate recognition of this disease. Nail dystrophy, including onycholysis, pitting, and hyperkeratosis, is included in the CASPAR criteria [5]. Nail involvement is usually present in psoriatic disease, and the fingernails are more commonly affected than the toenails. In clinical examinations, high-frequency sonography provides valuable information regarding structural changes in the nail unit in PsO patients [6]. These morphostructural changes in PsO and PsA patients include ventral nail plate deposits, increased nail plate thickness, and irregular or completely fused nail plates [7]. Arbault et al. reported that feasibility and reliability were satisfactory for ultrasonography (US) of the nail in PsA [8].

In addition to nail changes, US can accurately detect enthesis involvement, including thickened tendons, hypoechogenicity, bursitis, and bone erosion [9]. The Glasgow Ultrasound Enthesitis Scoring System (GUESS) score is significantly higher in patients with PsO than in healthy controls [10,11]. In addition, entheseal involvement has been reported in asymptomatic patients with PsO, with a high prevalence of subclinical enthesopathy [11]. US may be useful for rheumatologists and dermatologists to identify subclinical PsA based on identification of enthesopathy in the distal interphalangeal (DIP) joint, quadriceps tendon, and the distal and proximal patellar ligament.

In our study, we aimed to establish the status of ultrasound as a valuable tool to evaluate nail changes and enthesopathy in psoriatic patients. This cross-sectional study revealed clinical characteristics and ultrasonographic differences between PsO and PsA in the nails and enthesis.

## 2. Materials and Methods

### 2.1. Study Design

This single-centre, cross-sectional study was performed between September 2018 and January 2019 at Chang-Gung Memorial Hospital Linkou branch, Taiwan. The study population consisted of 189 non-consecutive patients comprising 35 with PsO and 154 with PsA (CASPAR criteria) attending the outpatient clinics of the Dermatology and Rheumatology units at Chang-Gung Memorial Hospital. The exclusion criteria were age <18 years, mycosis, trauma and/or local corticosteroid injection within the past 6 weeks at the DIP joints and/or lower-limb entheses. Patients with other causes of hand or leg enthesopathy were also excluded, for example, rheumatoid arthritis, osteoarthritis, or crystalline deposition disease. The study was conducted in accordance with the tenets of the Declaration of Helsinki and local regulations. Ethical approval for the study was obtained from the Chang-Gung Memorial Hospital Local Ethics Committee, and all patients provided informed consent.

The following parameters were included: age, sex ratio, disease duration (years), family history, uveitis, body mass index (BMI), nail PsO, medical treatment including classical disease-modifying antirheumatic drugs (cDMARDs; methotrexate, sulfasalazine, leflunomide, cyclosporine), tumour necrosis factor inhibitor (certolizumab, golimumab, adalimumab, etanercept), interleukin-17 inhibitor (ixekizumab, secukinumab), interleukin-12 and interleukin-23 inhibitor (ustekinumab), and interleukin-23 inhibitor (guselkumab).

### 2.2. Ultrasound Assessment

Real-time high-resolution US was performed by an experienced rheumatologist using a Siemens ACUSON P300™ US system equipped with a variable-frequency transducer ranging from 6 to 18 MHz (focal range 0.2–3 cm, image field 16 mm), to observe nail anatomy. The parameters for B-mode US examinations were set to maximise the precision of detection. The settings for structures were: full nail structure (18 Hz), superior pole of the patella—quadriceps tendon enthesis (16 Hz), inferior pole of the patella—proximal patellar ligament enthesis (16 Hz), tibial tuberosity—distal patellar ligament enthesis (16 Hz), superior pole of the calcaneus—Achilles tendon enthesis (16 Hz), inferior pole of the calcaneus—plantar aponeurosis enthesis (6 Hz). Definition for abnormal enthesopathy thickness: quadriceps tendon > 6.1 mm, proximal and distal patellar ligament > 4 mm, Achilles tendon > 5.29 mm, plantar aponeurosis > 4.4 mm.

The patient was asked to sit with the forearm in a neutral position over a table, and the nails were scanned in longitudinal and transverse planes. We ensured a sufficient amount of US gel was applied to avoid alteration of nail thickness caused by transducer compression. Fingernails were scanned in grayscale mode with an 18 MHz transducer, to detect morphostructural changes (nail plate thickness, Wortsman’s classification, nail bed thickness, hyperechoic spots, nail matrix thickness or loss) [12]. In more detail, nail plate thickness was determined by measuring the distance between the ventral and dorsal nail plates, and nail bed thickness defined as the distance from the ventral nail plate to the dorsal side of the distal phalange 2.5 mm from the proximal nail fold. The following elemental lesions were evaluated by grayscale US assessment according to the Brown University Nail Enthesis Scale (BUNES) preliminary definition of onychopathy (0–90): Wortsman’s classification of the nail plate, thickness of the nail bed, hyperechoic spots on the nail bed, and nail matrix loss [7]. The nail plate, nail bed, and nail matrix thickness were measured in millimetres.

Abnormally hypoechoic and/or thickened tendons at the bony attachment, and bony changes including thickness, bursitis, enthesophytes, and erosions, were evaluated by grayscale US assessment at the quadriceps tendon enthesis, proximal patellar ligament enthesis, distal patellar ligament enthesis, Achilles tendon enthesis, and plantar aponeurosis enthesis, according to the GUESS (0–36) preliminary definition of enthesopathy [13].

### 2.3. Nail Change Assessment

The nail psoriasis severity index (NAPSI) was used for severity grading, which included parameters of the nail matrix (pitting, leuconychia, red spots in the lunula, crumbling) and nail bed (onycholysis, splinter haemorrhage, subungual hyperkeratosis, oil stains).

### 2.4. Statistical Analysis

Statistical analyses were performed using IBM SPSS Statistics 25.0 for Windows (IBM, Armonk, NY, USA). The results were evaluated using descriptive statistics. Linear correlations were represented using Pearson’s correlation coefficient (r). Continuous variables were expressed as the mean ± standard deviation depending on the distribution, and categorical variables were expressed as percentages with the corresponding 95% confidence interval (95% CI). The independent-samples t-test was applied to compare continuous variables, which were expressed as frequencies and crosstabs. Pearson’s Chi-square test was applied to compare categorical variables. Associative US features with the presence of nail involvement were also calculated using binary logistic regression analysis, adjusted for other variables. Inferential statistical analysis was conducted at a significance level of 5%, and *p* < 0.05 was taken to indicate statistical significance.

## 3. Results

### 3.1. Patient Characteristics

The study population consisted of a total of 189 patients comprising 154 with PsA and 35 with PsO. The characteristics of the study population are presented in Table 1. Nail PsO was observed in 158 of the 189 patients (83.6%).

The PsA and PsO groups showed significant differences in mean age (49.3 ± 14.0 vs. 43.5 ± 14.6, respectively; *p* = 0.028), disease duration (12.96 ± 11.14 vs. 8.83 ± 9.33; *p* = 0.043), GUESS value (3.9 ± 2.6 vs. 2.8 ± 1.8; *p* = 0.004), and NAPSI value (21.4 ± 21.4 vs. 33.9 ± 27.5; *p* = 0.016). The PsA and PsO groups showed a modest linear correlation between NAPSI and BUNES scores (r = 0.365; *p* < 0.0001). The rates of administration of sulfasalazine, leflunomide, and adalimumab were significantly higher in the PsA group than the PsO group (49 of 154 vs. 1 of 35, respectively; *p* = 0.001; 27 of 154 vs. 1 of 35; *p* = 0.031; 16 of 154 vs. 0 of 35; *p* = 0.050).

### 3.2. Ultrasonography Findings at Finger Nails

After excluding samples that were not available, 1880 fingernails in the total study population were evaluated: 1530 from the PsA group and 350 from the PsO group.

According to the morphostructural US (Figure 1A–C), there were no significant differences in nail thickness or nail morphometric changes between the PsA and PsO groups (Table 2). Across the total study population, the nail plate thickness was 0.44 ± 0.21 mm, nail bed thickness was 1.78 ± 0.53 mm, and nail matrix thickness was 1.96 ± 0.47 mm. Moreover, 783 of 1880 (41.6%) fingernails showed normal morphometry (Wortsman type 0) and 1097 of 1880 (58.4%) fingernails showed abnormal morphometry (Wortsman types 1–4) in the nail plate. Furthermore, 1355 of the 1880 fingernails showed thickening of the nail bed, 30 (1.6%) showed hyperechoic spots, and 1351 (71.9%) showed nail matrix loss.

The patients with psoriatic disease and clinical fingernail involvement presented onychopathic signs: pitting, leuconychia, crumbling, onycholysis, splinter haemorrhage, subungual hyperkeratosis, and oil stains (Appendix A). Subungual hyperkeratosis was significantly more common in the PsO group than the PsA group (25.7% vs. 11.7%, respectively; *p* = 0.035), while there were no significant differences in the other subtypes between the two groups.

Binary logistic regression analysis indicated clinical features associated with fingernail thickness (Appendix A). In particular, NAPSI (OR = 0.977; CI = 0.959–0.995; *p* = 0.013) and GUESS (OR = 1.332; CI = 1.046–1.696; *p* = 0.020) scores were significantly associated with fingernail thickness, independent of age, sex, and BMI.

### 3.3. Ultrasonographic Findings at the Lower-Limb Enthesis

Multiple sites of enthesopathy were detected with clinical lower-limb involvement in PsA and PsO patients, including quadriceps tendon enthesis, proximal patellar ligament enthesis, distal patellar ligament enthesis, Achilles tendon enthesis, and plantar aponeurosis enthesis (Table 3). US indicated abnormally hypoechoic lesions, thickened tendons at sites of bone attachment, and changes in bones. The ligaments of the lower limbs were thicker in the PsA group than the PsO group, but the differences between the two groups were not significant, except for distal patellar ligament thickness and Achilles tendon thickness, which were significantly greater in patients with PsA than in those with PsO (3.7 ± 0.7 mm vs. 3.4 ± 0.7 mm, respectively; *p* = 0.032; 4.8 ± 1.0 mm vs. 4.5 ± 0.8 mm; *p* = 0.044) (Table 3).

### 3.4. Uniform Score Systems under Different Systemic Therapies

Subgroup analysis was carried out to investigate whether treatments impacted psoriatic nail change and enthesopathy. We selected three uniform score systems (NAPSI, BUNES and GUESS) and defined them as six categorical variables (NAPSI ≥ 20, NPASI < 20, BUNES ≥ 22, BUNES < 22, BUESS ≥ 3, BUNESS < 3). Next, we compared six parameters in relation to psoriatic disease (PsA, PsO) and three subgroup therapies (only cDMARDs, only bDMARDs, and combined therapy). Under Pearson’s Chi-square analysis, there were significant differences between high or low GUESS values and PsO or PsA. (*p* = 0.030). High or low GUESS values also showed significant differences for the three systemic therapies (*p* = 0.021) (Table 4). Furthermore, for the high or low GUESS parameter, comparisons of cDMARDs versus bDMARDs and bDMARDs versus combined therapy revealed significant differences (*p* = 0.012, *p* = 0.007, respectively).

## 4. Discussion

In this study, we used ultrasonography to quantify the severity of enthesopathy affecting nails and lower extremities, aiming to reveal correlations between enthesopathy and subgroups of psoriatic diseases. The prevalence of enthesopathy involvement was demonstrated under designated ultrasound survey. PsA was associated with a higher GUESS value, and nail US was not different between PsO and PsA groups.

In our study population, higher proportions of nail plate, nail bed, and nail matrix abnormalities were seen in patients with PsO than in those with PsA. The mean nail plate thickness was greater in the PsA group, although the difference was not significant. Conversely, the mean nail bed and nail matrix thicknesses were greater in the PsO group, although the differences were not significant (Table 2). However, Naredo et al. reported that nail bed thickness, nail plate thickness, and B-mode scores were significantly higher in the nails of patients with PsO than those with PsA [14]. The nail imaging findings in our study were similar to those of previous studies by Aydin et al. and Idolazzi et al [15,16]. Those studies reported nail involvement in both PsA and PsO, with higher frequencies of structural and inflammatory changes in both groups compared with healthy controls, but no significant differences between the PsA and PsO groups [15,16]. The sample size of the PSO group was small. That may be the reason that that NAPSI value was higher in the PSO group, but no differences in nail morphometry parameters were found between the groups. Numerous surveys have shown a remarkably high frequency of clinically occult musculoskeletal symptoms in psoriasis patients, ultrasound in particular having revealed a high prevalence of subclinical enthesitis and other inflammatory changes. Strategies must recognize and incorporate assessment of community-based psoriasis sufferers with only mild skin disease, as this group is at particular risk of PsA [17].

Among our patients, the GUESS values were significantly higher in the PsA group than the PsO group. A previous study showed that common sites of enthesitis in PsA patients included the insertion of the quadriceps tendon into the patellar bone, insertion of the proximal patellar ligament into the patellar bone, insertion of the distal patellar ligament into the tibial tuberosity, insertion of the Achilles tendon into the calcaneus bone, and insertion of the plantar aponeurosis into the calcaneus bone [18]. Three studies have shown the validity of US for early detection of subclinical enthesopathy in psoriatic diseases [19,20,21]. 

In contrast to lesion thickness, the distal patellar ligament and Achilles tendon were significantly thicker in PsA than in PsO (Table 3). Moshrif et al. reported that thickening of the Achilles tendon was the most common sign in patients with enthesitis [22]. In the ULISSE study, Achilles tendon lesion was found to be significantly higher in PsA than in PsO [17]. However, this was contrary to the report by Michelsen et al. indicating no association between clinical and US signs of Achilles enthesitis in PsA [23]. Graceffa et al. reported that thickness of enthesis at the superior pole of the patella was significantly increased in PsA, but not in PsO nor healthy controls [24]. Macchioni et al. also reported that entheseal thickening of the Achilles tendon was significantly greater in PsA than PsO [25]. However, isolated peripheral enthesitis was not significant in most of our study population, except in cases of Achilles tendon thickness and distal patellar ligament thickness.

In our study, the rate of clinical nail abnormalities was higher in the PsO group than the PsA group (32 of 35 (91.4%) vs. 126 of 152 (82.9%), respectively). There were no significant differences between the two groups, except in nail bed subungual hyperkeratosis (Appendix A). Castellanos-González et al. and Naredo et al. reported that there was a significant correlation between target NAPSI value and evidence of enthesopathy [26]. Our observation that GUESS values were significantly higher in patients with PsO than in healthy controls or those with other dermopathies was consistent with the reports of Gutierrez et al. and Pistone et al. [11,27]. Both of those studies revealed the ability of US to detect signs of subclinical enthesopathy, and demonstrating its value in detecting entheseal changes in psoriatic patients according to GUESS rating.

In the present study, GUESS value was significantly higher in the PsA group than the PsO group on binary logistic regression analysis (OR = 1.332; CI = 1.046–1.696; *p* = 0.020) (Appendix A). In addition, there was a modest correlation between GUESS and BUNES scores in the PsA group. Ash et al. reported that enthesopathy and inflammation scores were higher in PsO patients with nail disease than in those without nail disease, or in healthy controls [28]. Furthermore, El Miedany et al. also reported that structural joint damage was significantly associated with onychopathy-defined changes in the trilaminar appearance of the nails, extensor tendon enthesopathy, or enhanced vascularity in the nail bed (OR = 2.30, 95% CI = 1.17–3.69) [29]. Those studies supported the suggestion that clinical evidence of onychopathy may be correlated with enthesopathy in psoriatic patients. Increased GUESS values and/or entheseal thickness are more common in patients with PsA than in those with PsO.

This study had some limitations. First, we did not investigate intraobserver reliability of US features. Second, there was a marked difference in the numbers of patients with PsA (*n* = 154) and those with PsO (*n* = 35). Third, we could not discriminate the directionality of the association between nail thickening and skin manifestations of psoriatic disease. Fourth, the sonographer was not blinded to the presence of enthesopathy and nail involvement. Furthermore, the longer duration of PsA might be a suspicious confounding factor in GUESS analysis. We had no data about the subgroups of PsA, the severity of PsA, nor the severity of PsO. Finally, treatment plans and disease activity, including remission, might be important confounding factors.

## 5. Conclusions

In conclusion, US was valuable for evaluating nail changes and enthesopathy in PsO. We found several quantitative parameters that may be useful in US assessment of psoriatic nails and the entheseal complex. Further US studies of nail and entheseal involvement in psoriatic diseases are required for application in clinical practice.

## Figures and Tables

**Figure 1 jcm-11-06296-f001:**
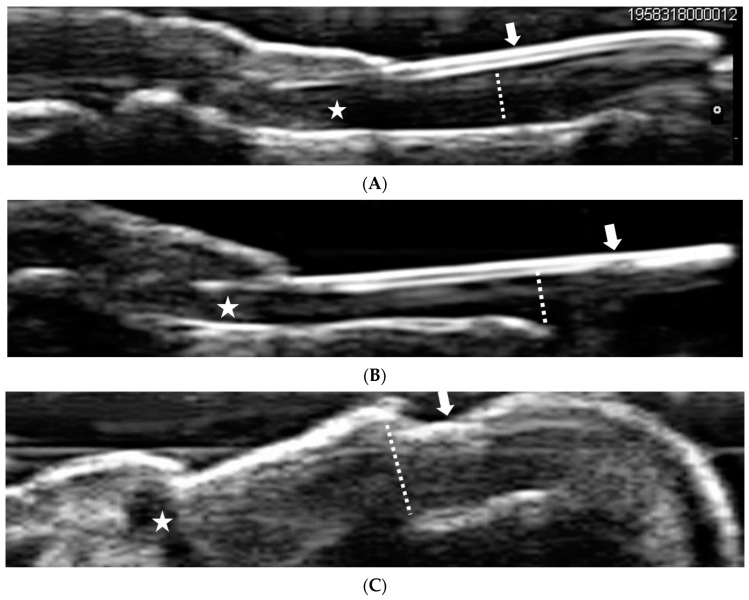
Comparison health control with nail psoriasis. (**A**). Grayscale ultrasonogaphy of the fingernail in a healthy control. Arrow: trilamellar and convex structure of nail plate with two hyperechoic lines, ventral and dorsal, separated by a hypoechoic line. Dashed line: the diameter of the nail bed is thinner in the distal part than the proximal. Asterisk: nail matrix present. (**B**). Grayscale ultrasonography of the fingernail in mild psoriatic nail disease. Arrow: interruption of hyperechoic line and loss of convex structure of nail plate (Wortsmann type II). Dashed line: loss of thinner distal part of nail bed. Asterisk: nail matrix present. (**C**). Grayscale ultrasonography of the fingernail in severe psoriatic nail disease. Arrow: a single and wavy hyperechoic layer losing its normal trilamillar appearance (Wortsmann type IV). Dashed line: thickening of nail bed. Asterisk: loss of normal nail matrix structure.

**Table 1 jcm-11-06296-t001:** Demographic profile of study population and disease-related data for PsO and PsA patients.

Parameters	All (*n* = 189)	PsA (*n* = 154, 81.5%)	PsO (*n* = 35, 18.5%)	*p* Value
Age (years), mean ± SD	48.3 ± 14.2	49.3 ± 14.0	43.5 ± 14.6	0.028
Male:female	112:77	90:64	22:13	0.631
Disease duration (years), mean ± SD	12.2 ± 10.9	12.96 ± 11.14	8.83 ± 9.33	0.043
Family history, N (%)	45 (23.8%)	34 (22%)	11 (31.4%)	0.241
Uveitis, N (%)	6 (3.2%)	6 (3.9%)	0 (0%)	0.342
Body mass index (kg/m^2^), mean ± SD	27.56 ± 5.272	27.86 ± 5.237	26.24 ± 5.312	0.136
Nail psoriasis, N (%)	158 (83.6%)	126 (82.9%)	32 (91.4%)	0.318
cDMARDs, N %	112 (59%)	97 (63.0%)	15 (42.9%)	0.182
Methotrexate	71 (37.6%)	58 (0.4%)	13 (0.4%)	0.916
Sulfasalazine	50 (26.5%)	49 (31.8%)	1 (2.9%)	0.001
Leflunomide	28 (14.8%)	27 (17.5%)	1 (2.9%)	0.031
Cyclosporine	6 (3.2%)	4 (2.6%)	2 (5.7%)	0.318
b-DMARDs, N%	47 (25%)	43 (27.9%)	4 (11.4%)	0.137
TNFi, N %	24 (12.7%)	23 (14.9%)	1 (2.9%)	0.059
Golimumab	2 (1.1%)	2 (1.3%)	0 (0%)	0.506
Adalimumab	16 (8.5%)	16 (10.4%)	0 (0%)	0.050
Etanercept	6 (3.2%)	5 (3.2%)	1 (2.9%)	0.934
IL17i, N %	19 (10%)	18 (11.7%)	1 (2.9%)	0.312
Ixekizumab	2 (1.1%)	2 (1.3%)	0 (0%)	0.506
Secukinumab	18 (9.5%)	17 (11.0%)	1 (2.9%)	0.149
IL12, 23i, N %	6 (3.2%)	4 (2.6%)	2 (5.7%)	0.313
Ustekinumab	6 (3.2%)	4 (2.6%)	2 (5.7%)	0.318
IL23i, N %	1 (0.5%)	1 (0.6%)	0 (0%)	0.639
Guselkumab	1 (0.5%)	0 (0%)	1 (0.6%)	0.638
NAPSI (points), mean ± SD	23.8 ± 23.1	21.4 ± 21.4	33.9 ± 27.5	0.016

PsO, psoriasis; PsA, psoriatic arthritis; *n*, the number of patients; cDMARD, conventional disease-modifying antirheumatic drugs; bDMARDs, biological disease-modifying antirheumatic drugs; TNFi, tumor necrosis factor inhibitor; IL17i, interleukin 17 inhibitor; IL12, 23i, interleukin 12, 23 inhibitor; IL23i, interleukin 23 inhibitor; GUESS, Glasgow Ultrasound Enthesitis Scoring System; NAPSI, nail psoriasis severity index; BUNES, Brown University Nail Enthesis Scale; SD, standard deviation.

**Table 2 jcm-11-06296-t002:** Nail morphometry in ultrasounds.

Parameters	All	PsA (*n* = 154)	PsO (*n* = 35)	*p* Value
BUNES (points), mean ± SD	22.9 ± 8.5	22.6 ± 8.6	24.3 ± 8.3	0.292
Nail plate	1097 (58.4%)	853 (55.8%)	244 (69.7%)	0.276
Normal	783 (41.6%)	677 (44.2%)	106 (30.3%)	0.291
Wortsman type 0	783 (41.6%)	677 (44.2%)	106 (30.3%)	0.291
Abnormal	1097 (58.4%)	853 (55.8%)	244 (69.7%)	0.276
Wortsman type 1	206 (11.0%)	166 (10.8%)	40 (11.4%)	0.355
Wortsman type 2	211 (11.2%)	167 (10.9%)	44 (12.6%)	0.414
Wortsman type 3	84 (4.5%)	65 (4.2%)	19 (5.4%)	0.388
Wortsman type 4	595 (31.6%)	455 (29.7%)	141 (40.3%)	0.279
Nail plate thickness, mean ± SD	0.44 ± 0.21 mm	0.45 ± 0.21 mm	0.40 ± 0.25 mm	0.165
Nail bed	1362 (72.4%)	1083 (70.8%)	279 (79.7%)	0.631
Thickness	1355 (72.1%)	1077 (70.4%)	278 (79.4%)	0.604
Hyperechoic spotting	30 (1.6%)	24 (1.6%)	6 (1.7%)	0.670
Nail bed thickness, mean ± SD	1.78 ± 0.53 mm	1.73 ± 0.45 mm	1.98 ± 0.77 mm	0.072
Nail matrix	1351 (71.9%)	1084 (70.8%)	267 (76.3%)	0.551
Loss	1351 (71.9%)	1084 (70.8%)	267 (76.3%)	0.551
Nail matrix thickness, mean ± SD	1.96 ± 0.47 mm	1.94 ± 0.45 mm	2.04 ± 0.54 mm	0.270

PsO, psoriasis; PsA, psoriatic arthritis; *n*, the number of patients.

**Table 3 jcm-11-06296-t003:** GUESS values and tendon thickness of PsA and PsO patients.

Parameters	All (*n* = 189)	PsA (*n* = 154)	PsO (*n* = 35)	*p* Value
GUESS (points), mean ± SD	3.7 ± 2.5	3.9 ± 2.6	2.8 ± 1.8	0.004
Superior pole of the patella—quadriceps tendon enthesis	75 (39.7%)	64 (41.6%)	11 (31.4%)	0.588
• Quadriceps tendon thickness > 6.1 mm	61 (32.3%)	53 (34.4%)	8 (22.9%)	0.303
Quadriceps tendon thickness (mm), mean ± SD	5.4 ± 1.0 mm	5.4 ± 1.0 mm	5.1 ± 1.1 mm	0.056
• Suprapatellar bursitis	1 (0.5%)	1 (0.6%)	0 (0%)	0.622
• Superior pole of patella erosion	0 (0%)	0 (0%)	0 (0%)	-
• Superior pole of patella enthesophyte	22 (11.6%)	19 (12.3%)	3 (8.6%)	0.159
Inferior pole of the patella—proximal patellar ligament enthesis	130 (68.8%)	104 (67.5%)	26 (74.3%)	0.636
• Patellar ligament thickness > 4 mm	129 (68.3%)	103 (66.9%)	26 (74.3%)	0.626
Proximal patellar ligament thickness (mm), mean ± SD	4.4 ± 0.9 mm	4.5 ± 0.9 mm	4.2 ± 0.8 mm	0.160
• Inferior pole of patella erosion	4 (2.1%)	4 (2.6%)	0 (0%)	0.320
• Inferior pole of patella enthesophyte	2 (1.1%)	1 (0.6%)	1 (2.9%)	0.272
Tibial tuberosity—distal patellar ligament enthesis	94 (49.7%)	82 (53.2%)	12 (34.3%)	0.364
• Patellar ligament thickness > 4 mm	72 (38.1%)	63 (40.9%)	9 (25.7%)	0.140
Distal patellar ligament thickness (mm), mean ± SD	3.6 ± 0.7 mm	3.7 ± 0.7 mm	3.4 ± 0.7 mm	**0.032**
• Infrapatellar bursitis	24 (12.7%)	21 (13.6%)	3 (8.6%)	0.438
• Tibial tuberosity erosion	8 (4.2%)	8 (5.2%)	0 (0%)	0.155
• Tibial tuberosity enthesophyte	9 (4.8%)	9 (5.8%)	0 (0%)	0.319
Superior pole of the calcaneus—Achilles tendon enthesis	95 (50.3%)	79 (51.3%)	16 (45.7%)	0.251
• Achilles tendon thickness > 5.29 mm	69 (36.5%)	57 (37.0%)	12 (34.3%)	0.251
Achilles tendon thickness (mm), mean ± SD	4.8 ± 1.0 mm	4.8 ± 1.0 mm	4.5 ± 0.8 mm	0.044
• Retrocalcaneal bursitis	7 (3.7%)	5 (3.2%)	2 (5.7%)	0.123
• Posterior pole of calcaneus erosion	10 (5.3%)	10 (6.5%)	0 (0%)	0.278
• Posterior pole of calcaneus enthesophyte	39 (20.6%)	35 (22.7%)	4 (11.4%)	0.145
Inferior pole of the calcaneus—plantar aponeurosis enthesis	32 (16.9%)	27 (17.5%)	5 (14.3%)	0.381
Plantar aponeurosis thickness > 4.4 mm	31 (16.4%)	26 (16.9%)	5 (14.3%)	0.416
• Plantar aponeurosis thickness (mm), mean ± SD	3.5 ± 0.7 mm	3.5 ± 0.8 mm	3.4 ± 0.6 mm	0.639
• Inferior pole of calcaneus erosion	1 (0.5%)	1 (0.6%)	0 (0%)	0.622
• Inferior pole of calcaneus enthesophyte	1 (0.5%)	1 (0.6%)	0 (0%)	0.622

PsO, psoriasis; PsA, psoriatic arthritis; *n*, number of patients; GUESS, Glasgow Ultrasound Enthesitis Scoring System; SD, standard deviation.

**Table 4 jcm-11-06296-t004:** NAPSI, BUNES and GUESS values in patients with psoriatic disease receiving systemic therapy.

Parameters	PSO (*n* = 35)	PSA (*n* = 154)	*p* Value	cDMARDs (*n* = 94)	bDMARDs (*n* = 29)	Combined Therapy (*n* = 18)	*p* Value
High NAPSI (≥20)	21 (60%)	72 (46.8%)	0.157	44 (46.8%)	15 (51.7%)	7 (38.9%)	0.693
Low NAPSI (<20)	14 (40%)	82 (53.2%)	0.157	50 (53.2%)	14 (48.3%)	11 (61.1%)	0.693
High BUNES (≥22)	21 (60%)	74 (48.1%)	0.168	51 (54.3%)	16 (55.2%)	7 (38.9%)	0.416
Low BUNES (<22)	13 (37.1%)	78 (50.6%)	0.168	42 (44.7%)	12 (41.4%)	11 (61.1%)	0.416
High GUESS (≥3)	12 (34.3%)	82 (53.2%)	0.030	60 (63.8%)	26 (89.7%)	10 (55.6%)	0.021
Low GUESS (<3)	23 (65.7%)	68 (44.2%)	0.030	31 (33.0%)	3 (10.3%)	8 (44.4%)	0.021

PsO, psoriasis; PsA, psoriatic arthritis; *n*, number of patients; cDMARDs, only conventional disease-modifying antirheumatic drugs; bDMARDs, only biological disease-modifying antirheumatic drugs; Combined therapy, cDMARDs plus bDMARDs; GUESS, Glasgow Ultrasound Enthesitis Scoring System; NAPSI, nail psoriasis severity index; BUNES, Brown University Nail Enthesis Scale.

## Data Availability

Restrictions apply to the availability of these data. Data was obtained from Chang Gung Memorial Hospital and are available from the authors with the permission of Chang Gung Memorial Hospital.

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
