# Peer review of "Ultrasound Can Be Usefully Integrated with the Clinical Assessment of Nail and Enthesis Involvement in Psoriasis and Psoriatic Arthritis"

_jcm, 2022, doi:10.3390/jcm11216296_

Round 1
Reviewer 1 Report
The present study was performed to examine the association between nail and enthesopathy detected by US in patients with PsO and PsA. I would like to know why authors wanted to make the research, what was the main aim and how was performed patients selection? All consecutive patients were included?
My main concern is the question what the study add to clinical practice - it should be referred clearly in the discussion.
Second - the association between nail and enthesopathy detected by US in patients with PsO and PsA can not be established with such patient selection and significant differences in biologic treatment between groups.
Overall I need more explanation what authors wanted to add to current knowledge
Author Response
Dear Reviewer,
We appreciate all of the valuable comments from the reviewers of our work. We have revised our manuscript, according to the reviewers’ comments, questions, and suggestions.
[Comment]
The present study was performed to examine the association between nail and enthesopathy detected by US in patients with PsO and PsA. I would like to know why authors wanted to make the research, what was the main aim and how was performed patients’ selection? All consecutive patients were included?
[Response]
In our study, we tried to establish the status of ultrasound as valuable tool to evaluate nail changes and enthesopathy in psoriatic patients. First goal is to quantify the severity of enthesopathy over nail and lower extremities with ultrasound. Second goal is to find the correlation between enthesopathy and subgroup of psoriatic diseases. Third goal is to find the prevalence of enthesopathy involvement under delegated ultrasound survey. We added related description in the introduction and discussion.
This is a cross-sectional study and no consecutive patient was included. We revised the methods as your suggestion.
[Comment]
My main concern is the question what the study add to clinical practice - it should be referred clearly in the discussion.
[Response]
In our study, we tried to establish the status of ultrasound as valuable tool to evaluate nail changes and enthesopathy in psoriatic patients. First goal is to quantify the severity of enthesopathy over nail and lower extremities with ultrasound. Second goal is to find the correlation between enthesopathy and subgroup of psoriatic diseases. Third goal is to find the prevalence of enthesopathy involvement under delegated ultrasound survey. We revised the discussion as your revision.
[Comment]
Second - the association between nail and enthesopathy detected by US in patients with PsO and PsA can not be established with such patient selection and significant differences in biologic treatment between groups.
[Response]
This was a cross-sectional study, it’s inevitable that most of patients had under clinical therapy. We made a subgroup analysis to investigate whether treatments impacted psoriatic nail change and enthesopathy. Under Pearson’s Chi-square analysis, we evaluate the difference in uniform score systems (high/low NAPSI, high/low BUNES and high/low GUESS) between 3 subgroup therapies(only cDMARDs, only bDMARDs and combine therapy). There was significantly difference in low and high GUESS. (p=0.021) Besides, in high/low GUESS parameter, cDMARDs versus bDMARDs and bDMARDs versus combine therapy revealed significantly difference. (p=0.012, p=0.007, separately) (Table 4) We believe that the association between lesion severity and biological treatment would be valuable for further investigation. We added related description in the result, discussion and limitation.
We look forward to hearing from you regarding our submission. We would be glad to respond to any further comments that you may have.
Sincerely, Authors
Reviewer 2 Report
Dear Authors,
I appreciate your paper, about the use of ultrasound in the assessment of nail and enthesis involvement in pateints affected by psoriasis and psoriatic arthritis
Even if the paper is of overall good quality, I suggest to clarify some points and eventually change some content in order to improve the presentation of your research
- In the paper both nails and entheses related to GUESS score are the object of the study. Since they are very different structures I think it will be correct to state the full parameters of the machine used for scan them. You may report all the settings for every structure (or group of structures if no change was made, such as the nail, the quadriceps tendon, the plantar fascia and so on)
- It's not clear in MATHERIALS AND METHODS section which are the inclusion or exclusion criteria. For example authors don't state if any condition like trauma at nail, other disease like micosys are mandatory for exclusion and how they considered them in patient evaluation. Furthermore, no data regarding disease activity are reported, so we don't know if the sample was in active disease, remission. We don't know also the percentage of disease activity/remission for every subpopulation. this is important to clarify if any bias due to the evident difference in sample size between PSA and PSO
- no data are reported about subset of disease in PSA group. The reader cannot discriminate if patients with PsA are affected by poliarticular or oligoarticular arthritis, if DIP are involved or not. Same happens for PSO group
- a crucial point for authors. In the paper the number of pathological nails found US findings despite clinical activity must be reported. it is a key point regarding your statement in RESULTS On binary logistic regression analysis, the clinical features associated with fingernail thickness. It comes along to ask you if you found not clinically affected nails but with increased thickness. This is very important since the NAPSI value is higher in PSO but no differences appear in any of US nail morphometry parameters between the groups. It would be essential to clarify this point since in DISCUSSION you state In our study population, higher proportions of nail plate, nail bed and nail matrix abnormalities were seen in patients with PsO than in those with PsA. The mean nail plate thickness was greater in the PsA group, although the difference was not significant. Conversely, the mean nail bed and nail matrix thicknesses were greater in the PsO group, although the differences were not significant. Since your data are consistent with those present in literature, your observations might be easily explained by the fact than clinically normal plates are, however, increased in thickness. You should check, if possible, this hypotesis. Moreover, regarding your statement With the exception of nail plate thickness, most of the nail changes observed clinically and on US may be more severe in PsO than PsA you should consider that all these aspects are also evident at US of the nail plate and the should create a strong difference between groups. May the sample size of PSO too little or the patients are mainly in remission?
- Regarding the comments about the GUESS and findings of entesophaty, reference 25, 31 and 26 are quite outdated. May be you can try to find more recent references for supporting this evidence? Consider data from ULISSE, as you correctly did, just for supporting this task
- the therapy is a strong confonding element. PSA patient has a relevant percentage treated with anti IL17 drugs and the majority of PSA patients receives bDMARD if compared with PSO patients. This may have alter the data, even in disfavour of sensibility and sensitivity of PSO group. I think you must include a strong statement (stronger than the one already present in the limitations) regarding this confonding factor.
- Another crucial point. I suggest to include in MATHERIALS and METHODS a careful evaluation of the sample size. You are comparing a parameter with very low variance (nail plate thickness) with others with a greater variance (quantitative parameters of the GUESS). It is mandatory to state which is the one you choose for calculating the sample size. This is not amendable for this study, since a great unbalance is present in the size of the two groups
I hope authors can reconcile all these points and only the crucial ones are mandatory, but also others are important to achive good quality for this paper.
Author Response
Dear Reviewer,
We appreciate all of the valuable comments from the reviewers of our work. We have revised our manuscript, according to the reviewers’ comments, questions, and suggestions.
[Comment]
In the paper both nails and entheses related to GUESS score are the object of the study. Since they are very different structures I think it will be correct to state the full parameters of the machine used for scan them. You may report all the settings for every structure (or group of structures if no change was made, such as the nail, the quadriceps tendon, the plantar fascia and so on)
[Response]
Real-time high-resolution US was performed by an experienced rheumatologist using a Siemens ACUSON P300™ US system equipped with a variable-frequency transducer ranging from 6 to 18 MHz (focal range 0.2–3 cm, image field 16 mm) to observe nail anatomy and enthesopathy. The parameters for B mode US examinations were set to maximize the precision of detection. The settings for structures were as below: All nail structure (18Hz), Superior pole of the patella—quadriceps tendon enthesis (16Hz), Inferior pole of the patella—proximal patellar ligament enthesis (16Hz), Tibial tuberosity—distal patellar ligament enthesis (16Hz), Superior pole of the calcaneus—Achilles tendon enthesis (16Hz), Inferior pole of the calcaneus—plantar aponeurosis enthesis (6Hz). Definition for abnormal enthesopathy thickness: quadriceps tendon > 6.1 mm, proximal and distal patellar ligament > 4 mm, Achilles tendon > 5.29 mm, plantar aponeurosis > 4.4 mm. We revised the methods as your revision.
[Comment]
It's not clear in MATHERIALS AND METHODS section which are the inclusion or exclusion criteria. For example authors don't state if any condition like trauma at nail, other disease like micosys are mandatory for exclusion and how they considered them in patient evaluation.
[Response]
We revised the methods as your revision. “The exclusion criteria were age < 18 years, mycosis, with trauma and/or local corticosteroid injection within the past 6 weeks at the fingernails and/ or lower limb entheses. In case of lesions interference, the patients with other causes of hands or legs enthesopathy were also excluded, for example, rheumatoid arthritis, osteoarthritis and crystalline deposition disease.”
[Comment]
Furthermore, no data regarding disease activity are reported, so we don't know if the sample was in active disease, remission. We don't know also the percentage of disease activity/remission for every subpopulation. this is important to clarify if any bias due to the evident difference in sample size between PSA and PSO
[Response]
Treatment plan and disease activity/remission might be the important confounding factors. Unfortunately, we do not have the data to revise the study. We added related description in the limitation.
[Comment]
No data are reported about subsets of disease in PSA group. The reader cannot discriminate if patients with PsA are affected by polyarticular or oligoarticular arthritis, if DIP are involved or not. Same happens for PSO group.
[Response]
PsA was grouped into five subsets: distal interphalangeal (DIP) predominant, symmetrical polyarthritis, asymmetrical oligoarthritis and monoarthritis, predominant spondylitis, and arthritis mutilans. Unfortunately, we do not have the data to revise the study. We added related description in the limitation.
[Comment]
A crucial point for authors. In the paper the number of pathological nails found US findings despite clinical activity must be reported. it is a key point regarding your statement in RESULTS On binary logistic regression analysis, the clinical features associated with fingernail thickness. It comes along to ask you if you found not clinically affected nails but with increased thickness. This is very important since the NAPSI value is higher in PSO but no differences appear in any of US nail morphometry parameters between the groups. It would be essential to clarify this point since in DISCUSSION you state In our study population, higher proportions of nail plate, nail bed and nail matrix abnormalities were seen in patients with PsO than in those with PsA. The mean nail plate thickness was greater in the PsA group, although the difference was not significant. Conversely, the mean nail bed and nail matrix thicknesses were greater in the PsO group, although the differences were not significant. Since your data are consistent with those present in literature, your observations might be easily explained by the fact than clinically normal plates are, however, increased in thickness. You should check, if possible, this hypotesis. Moreover, regarding your statement With the exception of nail plate thickness, most of the nail changes observed clinically and on US MAY BE MORE SEVERE IN PSO THAN PSA you should consider that all these aspects are also evident at US of the nail plate and the should create a strong difference between groups. May the sample size of PSO too little or the patients are mainly in remission?
[Response]
Actually, the nail plate/bed/matrix thickness is present in clinically normal nails in PsO and PsA. Besides, the sample size of PSO is small. The above may be the reasons that NAPSI value is higher in PSO but no differences appear in any of US nail morphometry parameters between the groups. We added related description in the discussion. The remission condition is uncertain due to the lack of disease activity evaluation such as PASI or DAPSA score. We will describe it in the discussion.
[Comment]
- Regarding the comments about the GUESS and findings of entesophaty, reference 25, 31 and 26 are quite outdated. May be you can try to find more recent references for supporting this evidence? Consider data from ULISSE, as you correctly did, just for supporting this task.
[Response]
We had already revised the new references 25, 31 and 26 in the discussion.
[Comment]
The therapy is a strong confonding element. PSA patient has a relevant percentage treated with anti IL17 drugs and the majority of PSA patients receives bDMARD if compared with PSO patients. This may have alter the data, even in disfavour of sensibility and sensitivity of PSO group. I think you must include a strong statement (stronger than the one already present in the limitations) regarding this confonding factor.
[Response]
This was a cross-sectional study, it’s inevitable that most of patients had under clinical therapy. We made a subgroup analysis to investigate whether treatments impacted psoriatic nail change and enthesopathy. Under Pearson’s Chi-square analysis, we evaluate the difference in uniform score systems(high/low NAPSI, high/low BUNES and high/low GUESS) between 3 subgroup therapies(only cDMARDs, only bDMARDs and combine therapy). There was significantly difference in low and high GUESS. (p=0.021) Besides, in high/low GUESS parameter, cDMARDs versus bDMARDs and bDMARDs versus combine therapy revealed significantly difference.(p=0.012, p=0.007, separately) (Table 4) In another study, Bektaş et.al reported that higher MDA(minimal disease activity) frequency was associated with continuation of first b-DMARD (OR:13.9 p<0.001). Acostal Felquer et.al also reported that treatment with bDMARDs in patients with PsO may reduce the risk of PsA development. We believe that the association between lesion severity and biological treatment would be valuable for further investigation. We added related description in the result, discussion and limitation.
[Comment]
Another crucial point. I suggest to include in MATHERIALS and METHODS a careful evaluation of the sample size. You are comparing a parameter with very low variance (nail plate thickness) with others with a greater variance (quantitative parameters of the GUESS). It is mandatory to state which is the one you choose for calculating the sample size. This is not amendable for this study, since a great unbalance is present in the size of the two groups.
[Response]
We added the nail plate/bed/matrix thickness and quantitative parameters of the GUESS in the materials and methods according to your suggestion.
We look forward to hearing from you regarding our submission. We would be glad to respond to any further comments that you may have.
Sincerely, Authors
Reviewer 3 Report
The manuscript titled Ultrasound is useful to integrate the clinical assessment of nail and enthesis involvement in psoriasis and psoriatic arthritis is interesting. It describes the comparison of the findings of nail and enthesis ultrasonography in patients with psoriasis and psoriatic arthritis. The Authors did not find significant differences between the analyzed groups in the PASI score and GUESS score. They concluded that the findings of nail USG are more severe in psoriasis and the USG findings of enthesopathy of the lower limb are more severe in psoriatic arthritis.
1. The course of the disease was longer in the group of patients with PsA than PsO. Can it influence some outcomes – for example, the GUESS score? Can the longer duration of PsA be a confounding factor in the presented analysis?
2. Did you notice the differences in NAPSI, BUNES, or GUESS scores between the group using conventional DMARDs compared to biological DMARDs? Such analysis can give an interesting insight into the treatment of options of PsA and PsO.
If the authors consider the above comments, I believe the work is valuable and can be published in the Journal of Clinical Medicine.
Author Response
Dear Reviewer,
We appreciate all of the valuable comments from the reviewers of our work. We have revised our manuscript, according to the reviewers’ comments, questions, and suggestions.
[Comment]
The course of the disease was longer in the group of patients with PsA than PsO. Can it influence some outcomes – for example, the GUESS score? Can the longer duration of PsA be a confounding factor in the presented analysis?
[Response]
The longer duration of PsA might be a suspicious confounding factor in GUESS score analysis. We added related description in the discussion and limitation.
[Comment]
Did you notice the differences in NAPSI, BUNES, or GUESS scores between the group using conventional DMARDs compared to biological DMARDs? Such analysis can give an interesting insight into the treatment of options of PsA and PsO.
[Response]
We made a subgroup analysis to investigate whether treatments impacted psoriatic nail change and enthesopathy. Under Pearson’s Chi-square analysis, we evaluate the difference in uniform score systems(high/low NAPSI, high/low BUNES and high/low GUESS) between 3 subgroup therapies(only cDMARDs, only bDMARDs and combine therapy). There was significantly difference in low and high GUESS. (p=0.021) Besides, in high/low GUESS parameter, cDMARDs versus bDMARDs and bDMARDs versus combine therapy revealed significantly difference.(p=0.012, p=0.007, separately) (Table 4) We believe that the association between lesion severity and treatment options would be valuable for further investigation. We added related description in the result and discussion.
We look forward to hearing from you regarding our submission. We would be glad to respond to any further comments that you may have.
Sincerely, Authors
Round 2
Reviewer 1 Report
Thank you for your response. I am satisfied with corrections
Reviewer 2 Report
Dear Authors,
I liked the improvement in the paper and major issues were almost amended. I think there are minor issues that still remain, such as the ones already I pointed as important, but I think your work can add some information and rise interest in this subject. Try to improve those issues for future studies and thanks for your work in this field of research